# "Christ for You and Me": A Lutheran Theology of Proclamation and the Presence of the Preacher

Samantha Gilmore

Ministry Division, Wartburg Theological Seminary, Dubuque, IA 52003, USA; sgilmore@wartburgseminary.edu

**Abstract:** This article asserts that the particularly personal nature of Lutheran preaching compels preachers to be personally present with the words they proclaim and the people in front of them as they give them the gift of the gospel. Lutheran preaching prioritizes and drives toward the explicit proclamation of Christ crucified that declares Christ's promises to be true "for you." This is personal. The gospel is proclaimed not in general for all, but for each. The preacher gives the gospel to the people in front of them, just as the bread and cup are given to the people in Holy Communion. Such an intimate task invites an intimate presence. After unpacking the term "personally present," this essay outlines three interrelated elements of Lutheran preaching that reveal the importance of the presence of the preacher, and a few words are offered to suggest ways in which these learnings may be of interest to those of other traditions. Finally, four exercises are provided to help homileticians foster this presence in their students and to assist preachers in fostering this presence during their preaching preparation.

**Keywords:** Lutheran preaching; performance studies; presence; proclamation; sacramental preaching; embodiment





## 1. Introduction

As a homiletician formed by Princeton Theological Seminary's department of Speech Communication in Ministry[1] and Kristin Linklater's embodied method of freeing the natural voice,[2] I have for many years been committed to cultivating the presence of the preacher. The same Word of God that revealed Godself through the incarnation reveals Godself through the proclamation of human bodies and voices today. Fostering the presence of the preacher is a way of honoring this very human, person-to-person way that God chooses to "save those who believe" (1 Cor 1:21). It is a way of faithfully attending to the sermon manuscript of the body, which is seen and communicates immediately and deeply to the congregation, with the same care that we give to the sermon manuscript of the page, which is rarely seen by anyone but the preacher.

In July 2022, I became the Assistant Professor of Homiletics at Wartburg Theological Seminary in Dubuque, IA. Wartburg's distinctly Lutheran context has provided me, a believer shaped by an ecumenical background and training, a delightful dive into the riches of Lutheran proclamation that has not only supported but deepened my own convictions about fostering the preacher's presence. Lutheran preaching prioritizes and drives toward the explicit proclamation of Christ crucified. By the power of the Holy Spirit, the listeners may, as Article IV of the Augsburg Confession states, "receive forgiveness of sin and become righteous before God out of grace for Christ's sake through faith" (Kolb et al. 2000, pp. 38, 40) in the preaching moment. This is personal. The gospel is proclaimed not in general for all, but for each, "for you." The preacher gives the gospel to the people in front of them like the bread and cup are given to the people in Holy Communion. Such an intimate task invites an intimate presence in the pulpit.

In this article, I argue that the particularly personal nature of Lutheran preaching compels preachers to be personally present with the words they proclaim and the people

in front of them as they give them the gift of the gospel. After unpacking what I mean by "personally present," I will outline three interrelated elements of Lutheran preaching that reveal the importance of the presence of the preacher. Then, I will offer a few words to my ecumenical partners to suggest ways in which these learnings may be of interest in their own contexts. Finally, I will provide four exercises to help homileticians foster this presence in their students and to help preachers foster this presence in their preaching.

## 2. Unpacking "Personally Present"[3]

A general sense of what it means to be "personally present" can be gathered from how we experience others to be so (or not) in our everyday relationships. Imagine someone glued to their phone all through dinner rather than engaging with you. While they are technically present in the room, you would probably not experience them being personally present to you. Their body would be turned inward and downward toward their phone and not you, their eyes focused and invested in their phone and not you. Imagine someone, when you speak to them, immediately stopping what they are doing, turning their body toward you, and looking at you to respond and engage. You probably would experience them to be personally present to you. Their entire body has shifted from their own work to face you, showing a focus and investment in you.

The term "present" can seem a bit hazy (Hooke 2010, p. 29). Yet, when I ask others what it means, I have noticed a pattern. People tend to answer this question not with words, at least at first, but by straightening their posture, opening their arms and chest toward me, and seeing me, meeting my eyes. Their body knows what it means to be present and readily shows me by becoming more present to me. Perhaps this term is not as nebulous as sometimes thought. We are simply more accustomed to speaking it with the body's language. I will now seek to translate this language into words as it relates to preaching in order to unpack the term for the purposes of my thesis.

### 2.1. Personally Present to the Words

When a preacher is personally present, there is a sense that the preacher is invested in communicating the words they have prepared. The preacher physically experiences their words as they say them, letting them pass through the heart, gut, and flesh on the way out of the mouth, so that the whole person is preaching. Like a manuscript made flesh, visible and audible to the congregation, the preacher's whole body (through facial expressions, eye contact, gesture, posture, stance, size and speed of movement, etc.) and voice (through tone, pitch, volume, speed, silence, breath, laughter, etc.) communicate, manifesting the preacher's interpretation of the meaning and feeling of the words and what is at stake in those words in a way that the words cannot do on their own.[4] The words require this presence of the body; they have to be enfleshed to be communicated and understood in the same way that a piece of sheet music composed for a woodwind has to be played, and played with the whole instrument—not just the mouthpiece (Gross 2017, p. 48).[5] This has performative implications; however, preaching is not centrally about the performance. Homiletical praxis is not about finding physical and vocal variety or prowess to draw attention to the preacher. It is about the preacher taking the risk of claiming, owning, embodying the words, being personally present to the words to share them with the hearers in a way that allows the hearers not merely to receive but experience them (Hooke 2010, p. 12).

An alternative to this personal presence to the words is simply reciting words without letting them move the body and voice. In such rote recitation, it is as though the words move straight from page to mouth with nothing personal in between. Homiletician Jana Childers refers to preachers who do this as "reading machines, not preachers" (Childers 1998, p. 92). The result of such unengaged recital is preaching that does not fully convey the meaning of the words, with the result that it requires much more strenuous effort for the listeners to follow, fill in the gaps, decipher contradictions in communication,[6] and let the words drop into their own heart, gut, and flesh to discern whether and why the words

matter. The preacher is not personally present, so the listeners have to overcompensate for that lack. Reciting is preaching that fails to be personally present to the words to communicate the meaning and feeling of the words and what is at stake within them, and in so doing, conveys that the words need not be taken too seriously. Regrettably, the listeners may receive this lack of seriousness with greater clarity than the words themselves.

*2.2. Personally Present to the People*

When a preacher is personally present, there is a sense that the preacher is in relationship with the listeners, that the preaching event is not unidirectional. The preacher is talking *to* the people rather than *at* them, and is speaking in their language, using examples and metaphors from their everyday lives so that the listeners experience the sermon and the gospel for them. Additionally, the preacher is affected by the listeners' presence. This is witnessed in the preacher's responsiveness to the listeners' participation in the preaching (e.g., listeners may nod, tilt their head, lean forward or back, raise an arm, clap, stand up, and other physical responses; they may hum or murmur in agreement, giggle, shout, gasp, and other verbal responses). The personally present preacher notices some of these things throughout the preaching event and responds in a natural way that lets the people know that they are seen and that their presence makes a difference. This can be as simple as smiling at the person giggling or responding to an energetically shouted "Preach!" by taking that energy on by raising the volume and picking up speed. In quieter and less overtly participatory contexts, simple eye contact that meets the energy expressed through the eyes of the person can communicate this presence.[7] These are not inauthentic responses. These are not things that the preacher looks for opportunities to do in order to indicate a relationship that is not actually there.[8] These are examples of basic communication that many people practice intuitively and effortlessly in everyday conversation to express that they are present with and appreciative of others. While it takes more preparation than everyday conversation (e.g., the preacher cannot be glued to their manuscript), it can also be practiced in the pulpit.

When this personal presence to the people is lacking, there can be a sense that the presence of the listeners makes no difference, that the listeners just happen to overhear the preaching rather than participate in it. There may be less of a sense that the sermon is for them personally. A listener may giggle or clap, but the preacher will keep talking as if they do not notice, as if the listener is not there. This behavior implies that the listeners, too, should pretend that what is happening is not happening. The experience can result in passivity on the part of the listeners. If they do not expect to be acknowledged and engaged, if they do not expect the preacher to be personally present, there is little reason for them to put forth the effort that personal presence requires on their side.

With this sense of the personal presence of the preacher to the words and people, we will turn to examine three interrelated elements of Lutheran preaching that reveal the importance of the personal presence of the preacher.

**3. Lutheran Preaching Compels the Preacher's Personal Presence**

The particularly personal nature of Lutheran preaching compels the personal presence of the preacher. This will be shown by unpacking the Lutheran understanding of God's Word as law and gospel, the prioritization of proclamation, and the sacramental sense of the gospel "for you."

*3.1. Law and Gospel*

Law and gospel are the ways God's Word is understood to function in Lutheran theology. Put more personally, law and gospel are two voices with which God's Word addresses human beings (Stjerna 2021, p. 77). Most basically, as law, God's Word reveals "the truth about the human condition," and as gospel, God's Word reveals "the truth about God in Christ" (Wengert 2013, pp. 32–33). Lutheran theology recognizes God using the law in three ways: (1) "to maintain order and restrain evil" (the "civil use of the law");

(2) "to reveal sin; to terrify the comfortable, self-satisfied person; and to put to death the old creature" in order to drive them toward faith in Christ (the "chief" and "theological use" of the law);[9] and (3) to provide instruction to believers in "Christian living" [(Kolb 2014, p. 170); the "pedagogical use" of the law (Hannan 2022, p. 47)].[10] God uses the gospel "to forgive and make alive" (Wengert 2013, p. 33).

Lutherans understand Christians to be simul iustus et peccator (simultaneously saint and sinner), both "already holy and righteous" before God in Christ and "not free from sin or sinning or the impact of all that it entails" (Stjerna 2021, p. 65). Thus, Christians need the voice of both law and gospel throughout their lives. The emphasis on and the volume of each are not to be applied indiscriminately, but according to "the historical context and the needs of the people" (Trelstad 2015, p. 214). Law and gospel are commonly written and presumed to be experienced in the same order, that is, first law and then gospel (Luther was adamant about this order). Lutheran theologian Marit Trelstad points out, however, that a practice like infant baptism suggests a gospel–law–gospel dynamic and that "the law only makes sense and only gains efficacy when rooted in the gospel which Jesus Christ both is and fulfills" (Trelstad 2015, pp. 215, 222). Lutheran historian Kirsi Stjerna also notes that given the weight of sin and humanity's "chronic failure to heed the divine guidance," the voice of the gospel that is not dependent on human action but on God's action in Christ is necessarily prioritized (Stjerna 2021, p. 65).

Lutheran preachers listen to the Word of God in Scripture with a particular interest in how God is working law and gospel on them and their people.[11] How is God's Word putting our old, idolatrous identity to death and giving us a new identity oriented toward trust in God as a child of God? What sin is being condemned, and how are redemption, forgiveness, and reconciliation offered? How are our stony hearts being revealed and destroyed and new hearts of flesh being given (cf. Ezek 36:26; Wengert 2013, p. 32)? How are the comfortable being afflicted and the afflicted comforted? Where are we lost and being found by God? God's Word as law and gospel never simply *means* something. It always *does* something to us (Wengert 2013, p. 32).

The death of the sinner and the resurrection of the saint are not merely beautiful ideas to talk about in preaching that the listener may experience at some other time. God works death and resurrection right then and there through the preacher's words in members of the congregation.[12] Death and resurrection are not happening in general but personally to the individuals to whom the preacher is preaching.[13] Just as pastors are compelled to be personally present for other moments of death and new life (such as baptisms, weddings, and funerals), so I propose that the preacher is compelled to be personally present for the law and gospel that God is working through the preacher's words—not because God's Word cannot function otherwise, but as a way of taking seriously what is happening. Lutheran preaching is not inconsequential. It is death and life, spiritually speaking.

### 3.2. Lutheran Preaching Prioritizes Proclamation

Lutheran preaching prioritizes proclamation. As Lutheran theologian Gerhard Forde (1927–2005) explains in his now-classic *Theology is for Proclamation*, preaching and proclamation are often used interchangeably; however, they are not the same. Proclamation is "the explicit declaration of good news, the gospel, the kerygma," which takes place both inside and outside of the pulpit, for example, in the sacraments (Forde 1990, p. 1). Proclamation is not talking *about* or explaining something that happened two thousand years ago. It is actively and directly declaring, ideally in the present tense from the first person to the second person, that the promises of the gospel of Christ crucified are promises "for you." Forde explains:

> The most apt paradigm for such speaking is the absolution "I declare unto you the gracious forgiveness of all your sins in the name of the Father, the Son, and the Holy Spirit." Proclamation is not "about" something other than itself. It does not point away from itself. It does not signify some other thing. It is the saying and the doing of the deed itself, for example, "*I* baptize *you*..." The deed is done,

unconditionally. It is not an account of what happened in the past, such as, "God so loved the world that he gave his only begotten Son," true as that is and, indeed, as much as it authorizes the primary discourse. Such accounts are past tense. Proclamation is present tense: I here and now give to you, Christ himself, the body and blood of the Savior. I do it in both Word and sacrament. This is God's present move, the current "mighty act" of the living God. (Forde 1990, p. 2; emphasis original)

As the preacher proclaims Christ and his benefits, God in Christ, by the power of the Holy Spirit, uses the preacher's words to give what they declare to those who believe. What is given is precisely what is declared: Christ and his benefits.[14] Lutheran theologian Fred W. Meuser (1923–2018) explains, "The proclaimed Word of God is not just preliminary to the sacraments, a lower stage of God's grace that we 'really' get through sacramental action. Rather, the apostolic message brings God and all God's gifts." Proclaiming is, in fact, God speaking: "Christian preaching, when it is faithful to the Word of God in the Scriptures about our need and God's response to it, is God speaking. When it presents Christ so that faith becomes possible, it is God speaking. It is God's very own audible address to all who hear it, just as surely as if Christ himself had spoken it" (Meuser 2006, pp. 136–37).

God's address is not merely speech. God's Word, more than mere words, "does what it says. It says what it does" (Bayer 2006, p. 76). As Lutheran theologian Robert Kolb (b. 1941) asserts: "[God's] Word in all its forms actually conveys and performs [God's] saving will. God designed [God's] Word in these forms as instruments of [God's] recreating power which accomplish what they announce. More than performative speech, they are creative speech, parallel to God's speaking in Genesis 1" (Kolb 2009, p. 132). When God speaks to believers, there is a new creation. This understanding is what allows Martin Luther (1483–1546) to write in the Large Catechism in reference to the proclamation that takes place during the Eucharist, "All those who let these words be addressed to them and believe that they are true have what the words declare" (Kolb et al. 2000, p. 470). This is the power of proclamation.

Preaching is harder to pin down than proclamation. We recognize the formal event that takes place during a worship service, traditionally from a pulpit by an ordained minister; however, the content is much broader. At any given moment, a preacher might be telling Bible stories, providing historical background on a Bible passage, teaching church doctrine, helping people think theologically about current issues and events, or calling people to put their faith into action. All of these things have an important place in preaching. None of these things, however, necessarily include or move toward proclamation; none guarantee a sermon that declares the gospel of Jesus Christ "for you" (Forde 1990, p. 1). A Bible story may be told entirely in the third person and the past tense; it is talking *about* something that happened to someone else at some other time and place. Talking *about* a current issue may feel quite distant and abstract to the listener if not done in a way that draws close the connections to contemporary experience; moreover, the focal issue may offer more bad news than good news. Calling people to action, if not done with care, may seem to have the stronger and more determinative word than the proclamation. An exhortation to action may even seem to contradict the proclaimed word, subtly suggesting that God's action in Christ has not accomplished all that is necessary for salvation and that human action is also essential.[15] Lutheran preaching prioritizes God's action in Christ through the Holy Spirit "for you" through proclamation.[16]

I assert that the priority of such a personal "I" to "you" present-tense declaration that, more than merely talking *about*, gives what it says, compels preachers to be personally present to those who are receiving this declaration. We assiduously attend to the content of our words, giving care even to the details of the person and tense in which we speak because of how it might exclude or include, push people away from or draw people near to the good news for them. We do this with the hope that God might be able to work with us more than God works in spite of us. The hope is the same with our attention to personal presence.

### 3.3. Lutheran Preaching Is "For You"

Some of the reason for the priority of proclamation is encapsulated in two little words that appear in Luther's writings and reveal the personal, faith-fostering, sacramental purpose of proclamation. The words are "for you." For example:

Therefore, when you begin to believe, you discover at the same time that everything in you is completely blameworthy, damnable sins, as Rom. 3[:23] states: "All have sinned and fall short of the glory of God." And Rom. 3[:10–12] says, "There is no one who is righteous," no one does good, "all have turned aside, altogether they have done worthless things." By this knowledge you will realize that you need Christ, who suffered and rose again *for you*, in order that, believing in him, you may become another human being by this faith, because all your sins are forgiven and you are justified by another's merits, namely, by Christ's alone. (Luther and Wengert 2016, p. 492; emphasis added)

The words "for you" reveal the profoundly personal nature of the gospel. Christ "suffered and rose again" not in a distant, abstract sense for all people but for each particular person who hears and believes the proclamation through which God in Christ speaks to them. It is true that Christ achieved forgiveness on the cross "once for all;" however, Luther explains, "he has not yet distributed or given it on the cross" (Luther 1958, pp. 213–14). He distributes and gives it through proclamation. Proclamation is prioritized in Lutheran preaching so that each hearer might not merely understand what God has done in Christ in general, but personally know and experience the benefits of this good news for themselves. Through proclamation, individuals become new human beings; individuals are forgiven and justified. Through proclamation, "for all" becomes "for you." Meuser goes so far as to say that without this proclamation, "without the word spoken by a believer, the Gospel cannot do its work" (Meuser 2006, p. 141).[17]

Another instance of Luther's "for you" highlights his primary goal for preaching as a whole: "Preaching, however, ought to serve this goal: that faith in Christ is promoted. Then he is not simply 'Christ' but 'Christ *for you* and me,' and what we say about him and call him affects us" (Luther and Wengert 2016, p. 508; emphasis added). Luther hopes that each person's faith in Christ might be strengthened through preaching, that each hearer might come to trust more completely in Christ. This sentiment is articulated more strongly in the Augsburg Confession, in which Article V proposes that preaching exists because God gifts faith in Christ through proclamation: "To obtain such faith God instituted the office of preaching, giving the gospel and the sacraments. Through these, as through means, he gives the Holy Spirit who produces faith, where and when he wills, in those who hear the gospel" (Kolb et al. 2000, p. 40). Proclamation is prioritized in Lutheran preaching because it is the reason Lutheran preaching exists. Whatever else may be done or discussed in the sermon, promoting Christ's trustworthiness so that individuals might have faith in him is the reason the preacher preaches.[18]

This eucharistic example of Luther's "for you" reveals his love for "the personal application of the gospel in the distribution of the sacraments" (Kolb 2009, p. 145):

Here stand the gracious and lovely words, "This is my body, given *FOR YOU*," "This is my blood, shed *FOR YOU* for the forgiveness of sins." These words, as I have said, are not preached to wood or stone but to you and me; otherwise, he might just as well have kept quiet and not instituted a sacrament. Ponder, then, and include yourself personally in the "YOU" so that he may not speak to you in vain. (Kolb et al. 2000, p. 473; emphasis added)

Luther encourages us to receive the Words of Institution as the needy and vulnerable human beings we are, that we might experience the for you-ness of the bread and cup and ponder it in our hearts (cf. Luke 2:19). These words, along with the promises of "forgiveness of sin, life, and salvation," (Kolb et al. 2000, p. 362) are ours.

Proclamation is prioritized in Lutheran preaching because of its own sacramental nature. Forde argues that the task is precisely the same:

> The preaching of the Word, that is, is to do the same thing as the sacrament—to give Christ and all his blessings. Indeed, since the Word is Christ, preaching is "pouring Christ into our ears" just as in the sacraments we are baptized into him, and he is poured into our mouths. (Forde 2017, p. 90)

The preacher declares what God has done for you in Christ, giving the gospel as freely to each as the bread and cup are placed in each one's hands, in faith that when the preacher does this, Christ himself speaks to grant forgiveness, life, and salvation to the hearers.[19]

A final example of these two words, taken up by Lutheran historian Timothy J. Wengert (b. 1950), reveals the importance of proclamation being personally communicated by one human being to others:

> Indeed, we would be better off replacing *sola Scriptura* with the phrase *solus Christus* (Christ alone) and, what amounts to the same thing, *solo Verbo*, by the Word alone—where "the Word" was for Luther not simply the Bible but its proclamation. Thus, already in 1522 Luther could write about the church that it is not a "quill house" but a "mouth house." God's Word was God's Word when proclaimed "*for you*," not when shut up in a book, where it was good only for others or for nothing. (Wengert 2013, p. 19; emphasis added)

We can read words on a page or scroll through them on a screen and choose to close them at any time, untouched. The experience is not the same as being presented with a word declared to be for us right now by someone who is personally present to us, looking at us, speaking to us directly. The prioritization of proclamation in Lutheran preaching offers a rare moment to simply receive the gift of "good news of great joy" (Luke 2:10), which is for us personally.[20]

The words "for you" all by themselves suggest a closeness, that someone has cared for you and drawn near to you to give you a gift. They reveal the sacramental nature of proclamation, which is the reason Lutheran preaching exists. Lutheran preaching seeks to foster the personal faith of those to whom the preacher preaches, to deepen their trust in Christ. This for you-ness compels a personal presence from preachers, not because the listeners' faith could not be fostered without it, but because it is a way of being that corresponds to such an intimate encounter.

This section outlining three interrelated elements of a Lutheran theology of preaching has shown that the particularly personal nature of Lutheran preaching compels preachers to be personally present with the words they proclaim and the people in front of them as they give them the gift of the gospel. Functioning as law and gospel, God's Word in Lutheran preaching is not satisfied to explain things or to remain at a distance. Instead, God's Word draws near and works the death of sinners and the resurrection of saints in the congregation as preachers preach. This time of condemnation and redemption, of affliction and consolation, of being lost and found, compels the preacher to be personally present as a pastor is present at other significant moments of death and new life. By prioritizing proclamation, Lutheran preaching compels preachers to speak to listeners directly, from the first to second person and in the present tense, as they declare the promises of the gospel to be for the listeners, in faith that all who believe will have what the preacher's words declare. This "I" to "you" giving of the gospel invites preachers to be as personally present with their listeners as they would be when they give other sacramental gifts to God's beloved. Through preachers' proclamation, God gives each believer life-overturning and transforming gifts: forgiveness, new life, and salvation. The sacramental nature of Lutheran preaching recognizes that Christ himself speaks when Christ's gospel is proclaimed, declaring all of its benefits "for you." These two little words encompass the personal way in which God in Christ, through the Holy Spirit, gives and fosters faith: through one human being communicating with others. God's personal choice compels a personal presence.

*3.4. A Note to My Ecumenical Partners*

Lutheran preaching is by no means the only preaching tradition that compels the preacher toward personal presence. My own background is ecumenical, and I have been thinking about the preacher's presence now for a number of years. The language of the Lutheran tradition has something beautiful and helpful to offer this discussion. God's Word functioning as law and gospel, as a Word that does not just *mean* but that *does* something to us, invites us to draw upon our own tradition's understanding of the Word's action in our preaching and to be present as the Word is present to witness that action. The prioritization of proclamation invites each believer to reflect upon the ways in which their own tradition summons them explicitly to give the gift of the gospel of Jesus Christ to their people through their preaching rather than talking *about* it. Lutheran sacramental theology also invites those of other traditions to ponder the ways in which a greater attentiveness to the present tense and first to second person might help them to be more present in that giving. The for you-ness of preaching invites all preachers to consider how our traditions may understand preaching in relation to giving our people the bread and the cup and how a sacramental, person-to-person embodiment of the gospel might translate to a more personal presence in our own embodied preaching. These points of connection are just a starting point for homileticians and preachers to adapt this offering to suit their own contexts.

**4. Fostering Personal Presence**

Lutheran preaching may compel preachers to be personally present, but putting this conviction into practice is often more easily said than done. In what follows, I offer four embodied exercises that I have used as a part of my own preaching preparation and have brought into the preaching classroom to help my students practice personal presence. Homileticians and preachers are encouraged to experiment with and modify them to suit their own contexts.

Theological education tends to be head-dominant, with the result that students who come into my classroom are surprised and sometimes uncomfortable with exercises like the ones I share here. Resistance from one or two students is not uncommon at the beginning of a semester; however, it does not usually last long. Openly and explicitly acknowledging that this kind of learning is probably unfamiliar and, as a result, may feel vulnerable to some people in the room goes a long way. Having a playful sense of humor about that can be immensely beneficial because laughter is relationship-building when it is kind.[21] Professors may wish to explain the purpose of an exercise before practicing it, particularly early on, when students may not trust them enough to jump in with both feet and experience the purpose for themselves.[22] Students can be invited to turn away from one another during exercises so that they feel less exposed. They may also feel more supported in what they are doing if the professor does an exercise with them as they lead it rather than merely providing instruction with words.[23]

Before I begin these exercises with a new group, I make sure students know that I do not want them to attempt anything that is not healthy or possible for their bodies to do. I invite them to talk to me one-on-one about physical limitations that may be helpful for me to keep in mind. This knowledge enables me to lead exercises in ways and with options that are the most beneficial for a particular group. I tell them that they can stop me any time an additional option for an exercise is needed, and to please immediately stop what they are doing if anything I ask them to do is causing them pain, so that we can find a pain-free path forward. I also note that imagining themselves doing the physical movements can be beneficial if, for any reason, actually doing them does not feel safe or possible.[24] They have agency over the way they choose to participate.[25]

*4.1. Embodying Faith: Remembering the Wisdom of the Body*[26]

This exercise is designed to foster the preacher's personal presence to the words, the content of what they proclaim, by (1) waking up the body (through which they experience

the words) in a very hands-on, tactile way, and then (2) inviting the body to remember and manifest moments when their own faith was fostered so that these experiences are rekindled and more readily felt when the preacher preaches. The preacher's connection to their own experience of the gospel enables them to be more personally present with the words through which they give it to others because they deeply know that of which they speak. Moreover, the gospel's goodness may draw them even closer to those to whom it is given as they look forward to the joy it will bring "for you."

I lead a class in this exercise; it can also be adapted for personal use. The words I choose here are inspired by a text my students read, *The Freedom of a Christian* (Luther and Wengert 2016). Locutions here can be replaced by words from other texts, if desirable in another pedagogical context.

1.  Vigorously pat down your entire body with your hands as if you are clapping all over your body, front and back, from your head to your toes. This invites energy and awareness to physical sensation.[27]
2.  Close your eyes for the rest of the exercise, as you feel comfortable, to help draw your focus inward.
3.  When was a time you felt nourished by the Gospel, fed by the Gospel? When was the last time you felt full of good news that was sustaining to you? If you have never felt this, imagine what it could be like to be fed, nourished, and sustained by the Gospel.

    a.  Once something has come to you, let your body find shape to express that feeling. Find a pose, a gesture, or a repeated movement that allows that feeling to become tangible in your body. (*Give time to find the shape.*) Take a deep breath in as you hold that shape. Say the word "nourished" aloud, full of the feeling of what your body is expressing. Let that go. (*Repeat this section with each word.*)

4.  When was a time you felt "justified" by grace through faith so that justification was palpable? When was a moment of recognition that in Christ, you, personally, have been made righteous before God, that in Christ, you are reconciled? If you have never felt this, imagine what it could be like to experience justification. (*Repeat 3.a.*)
5.  When was a time you felt "free"? When have you experienced freedom in Christ, freedom that frees you to turn outward and soften toward others? If you have never felt this, imagine what it could be like to know freedom in Christ in your body. (*Repeat 3.a.*)
6.  When was the last time you experienced "salvation," so that salvation was an event, you were brought from death to life, the old passed away, and the new came into being? Even if just for a moment, you could taste the kingdom of God. If you have never felt this, imagine what it could be like to see, hear, touch, taste salvation. (*Repeat 3.a.*)
7.  Martin Luther writes, "To preach Christ means to feed, justify, free, and save the soul—provided a person believes the preaching" (Luther and Wengert 2016, p. 491). At some moment, you believed the preaching, or you would not be here. Now you are the person who does the preaching. Thanks be to God.

*4.2. I Am Here in This Room with All of You*[28]

I use this exercise to foster the preacher's personal presence with other people. I initially have students practice it together as a group, which is how I describe it here. Students then practice a silent and swifter version in the pulpit in class immediately before they preach.

1.  Stand in a circle.[29]
2.  Close your eyes as you feel comfortable to help draw your focus inward.
3.  Turn your attention to the inside of your body. Scan through your body and notice any physical sensations that present themselves. Feel which parts of your feet are touching the floor. Feel the parts of your body that are being moved by your breath.

Feel your heart beating. Tune into your state of being. With that physical awareness, say aloud, "I am here…"

4.  Open your eyes and notice the room you are in with greater attentiveness than you usually do. See any artwork, cracked and faded paint, light coming in through the windows, the floor on which you are standing, the furniture, and anything else. Allow any memories or associations that come to you to be there. With that special awareness, say aloud, "…in this room…"

5.  See the people in the room with greater attentiveness than you usually do. Notice their facial expressions. Sense their presence, their energy, what their bodies are communicating. If your eyes meet someone else's, let yourself linger together for a moment. (*If the group is small enough, take the time to meet everyone eye to eye.*) Notice whether you are holding your breath as you see people, and keep breathing.[30] With that awareness of the people in the room, say aloud, "…with all of you."

6.  Keeping your eyes open and maintaining your connection to yourself, your space, and your group, say, "I am here in this room with all of you."

In a small group, have each person say it individually. Depending on the group and the particular person, you might have someone try again if the sentence does not ring true when they say it. Their attention may seem to be elsewhere so that they are not really "here." They may not be meeting the group eye to eye and taking them in, so they are not really "with all of you." They may speed through the whole sentence so quickly (to get it over with as quickly as possible) that none of it rings true. Remind them to keep breathing and to experience the truth of each word as they say it. It is vulnerable to say this sentence and to mean it, but that vulnerability leads to a personal presence that is palpable.

### *4.3. God Be in My Body[31]*

I have adapted, expanded, and added movement to the well-known "God be in my head" prayer attributed to the thirteenth-century English worship resource known as the Sarum Primer. In this exercise, I seek to foster the preacher's personal presence to the words they proclaim by fostering their presence to the God they proclaim through those words. I begin every class in my elective course, The Preaching Body, with this prayer practice. It can also be practiced individually by the preacher. As a part of preaching preparation, it supports preachers' personal presence as they listen to God's Word in Scripture and as they prepare to proclaim God's Word "for you."

1.  Stretch the right side of your body by keeping your right foot flat on the floor and stretching your right arm as high as it will go. Do the same on your left side.

2.  Stretch the right side of your body in the same way, this time yawning as you stretch, to open space for breath in the right side of the ribcage. Do the same on your left side.

3.  Bend your knees and round your spine forward into a "C" shape while bringing your arms forward as if holding a physioball. Yawn as you hold this stretch, to open space for breath in the back ribs.

4.  Bend your knees and reverse the movement, creating a backward "C" shape (or arch) with your spine while bringing your arms out to shoulder height. Yawn as you hold this stretch, to open space for breath in the front ribs and the heart.

5.  Release your head and neck forward and slowly undo your spine, vertebra by vertebra, as far down as provides a comfortable stretch for your spine. Let your knees be a little bent, and keep your weight evenly balanced on your feet. Yawn and feel your spine release a little further toward the ground on the outgoing breath.

6.  Slowly float your way back up to standing, taking a moment with just your head and neck still released to find your equilibrium, if needed.

7.  Bring your hands to your head and pray aloud: "God be in my head, and in my understanding."

8.  Bring your hands to your eyes and pray aloud: "God be in my eyes, and in my looking."

9.  Bring your hands to your ears and pray aloud: "God be in my ears, and in my hearing."
10. Bring your hands to your mouth and pray aloud: "God be in my mouth, and in my speaking."
11. Bring your hands to your heart and pray aloud: "God be in my heart, and in my pondering."
12. Bring your hands to your abdomen and pray aloud: "God be in my gut, and in my feeling."
13. Bring your hands to your legs and pray aloud: "God be in my legs, and in my moving."
14. Slowly release your spine down again, with the intention of releasing to God all the parts of you in which you have not been aware of God's presence.
15. As you float your way back up to standing, claim the truth that your body was "fearfully and wonderfully made" (Ps 139:14) by God. "Your body is a temple of the Holy Spirit" (1 Cor 6:19), a place in which God is pleased to dwell.
16. Bring your arms straight up toward the ceiling and pray aloud: God be in my body, and in my being.

*4.4. The Dimensionality of the Body*[32]

This longer exercise invites you to experience the power of your body to hold memory and meaning. It reveals the extent to which a deeper connection to your body helps deepen your personal presence to words, as ideas in your head become experiences in your body. This aids in the present-tense nature of proclamation. The preacher does not merely recite words that they thought of earlier, letting the congregation in on an experience and thought process now past, as this would relegate the congregation to the position of distant overhearers rather than participants. Rather, the preacher, by being personally present to the words, lives and embodies those words as they say them, so that they become a present-tense experience "for you."

1.  Inside–Outside

Start gently jogging in place. If possible, let the jog become a run and then a sprint in place. If possible, let the knees start to come up higher and higher toward the chest. Do this for about thirty seconds. Then, abruptly stop and close your eyes. Bring all of your attention **inside** your body. Feel all of the physical sensations that your body is communicating, such as the beating of your heart inside your chest, how your breath moves your body, the temperature of and perspiration on your skin. (*Do this twice.*)

Open your eyes. Bring all of your attention **outside** of your body until you don't feel anything; none of your awareness is on your physical sensations.[33] It may help to look at a focal point in the room or out a window.

Then, move your attention all the way **inside** of your body and all the way **outside** of your body a couple of times.

Play with percentages. Keep 90 percent of your awareness **inside** on your physical sensations, but move 10 percent of your awareness **outside** of yourself, then 75 percent **inside** and 25 percent **outside**, then 50 percent **inside** and 50 percent **outside**, then 25 percent **inside** and 75 percent **outside**, then 10 percent **inside** and 90 percent **outside**, and then move back to 50 percent **inside** and 50 percent **outside**. Consider as you do this what percentages feel most familiar, where you spend most of your time. Consider where you feel the most personally present. Consider what difference it might make if you preached from that place.

2.  Gravity–Levity

In this section of the exercise, you are free to sit, stand, lean, lie down, and move your body in any other way you would like to move in response to the instructions. There is no right way to do it. The goal is that the words become not merely ideas but experiences.

Moving through the space freely, let yourself become aware of the feeling of **gravity**. Let that feeling become dominant, taking over your body. Feel the heaviness of your feet; the heaviness of your legs, pelvis, torso; the heaviness of your arms and hands; the heaviness of your neck and head.

Let common phrases and metaphors become visceral in your body. (*Start with one and then add others when it feels like enough time has passed for the metaphor to have been explored. You might not get to all of them.*) Feel your head in the sand, the weight of the world on your shoulders, your heart drop inside your chest, a heavy heart, a rock in the pit of your stomach, the gravity of the situation, grounded, rooted. Let your body be moved by these words; let it do what it wants. Notice whether your body, or perhaps parts of your body, resists giving in to **gravity**. Notice any memories, texts, feelings that come to mind.

Now feel **gravity** being lifted away, up and out of your body, and let it be replaced totally with **levity**. Feel all of the places in you that are light, airy. Feel the lightness of your joints, the air in your lungs, perhaps your empty stomach. Fly, skip, gallop, leapfrog, dance, spin, levitate, tiptoe effortlessly. Float like an astronaut in space. Be still without any sense of collapse or stiffness. You are weightless.

Let common phrases and metaphors become visceral in your body. (*Start with one and then add others when it feels like enough time has passed for the metaphor to be explored. You might not get to all of them.*) Tangibly feel a weight lifted, lighter than air, as if you were walking on air, walking on sunshine, walking on eggshells. Tread lightly. Keep it light. Your head is in the clouds. Let your heart swell, soar, grow three sizes, leap outside of your chest. Feel butterflies in your stomach. The sky is the limit! Let your body be moved by these words; let it do what it wants. Notice whether your body, or perhaps parts of your body, resists giving in to **levity**. Notice any memories, texts, feelings that come to mind.

Now let **gravity** return, but without losing **levity**, so that you feel both equally. You feel the weight of **gravity**, but it does not pull you down. You feel the lightness of **levity**, but you do not float up to the ceiling. Notice whether your body is more receptive to one than the other. Notice any memories, texts, or feelings that come to mind. Consider where in the dimension of **gravity** and **levity** you feel the most personally present. Consider what difference it might make if you preached from that place.

3.    Front Body–Back Body

Walk through the space. Bring all of your attention and awareness to **the front of your body**; the front of your feet, shins, knees, thighs, pelvis, stomach, chest; the fronts of your arms, hands, throat, face. Feel the places that air touches **the front of your body**, the temperature of your skin. Feel the places your clothes touch **the front of your body**. Feel **the front of your body** leading you through the space. Notice the speed at which your body wishes to walk when led by this part of you, and let that happen; do not regulate yourself.

Let common phrases and metaphors become visceral in your body. (*Start with one and then add others when it feels like enough time has passed for the metaphor to be explored. You might not get to all of them.*) What are you looking forward to? What lies ahead? What would you like to do moving forward? Get ahead, press on. Face the music, face the facts. Feel yourself moving into your future. Notice whether you, or perhaps parts of you, resist coming all the way forward. Notice any memories, texts, or feelings that come to mind.

Now move all of your attention and awareness to **the back of your body**; the back of your feet, calves, knees, buttocks, lower back, upper back; the backs of your arms, hands, neck, head. Feel the places that air touches **the back of your body**, the temperature of your skin. Feel the places your clothes touch **the back of your body**. Imagine you had eyes back there. Feel **the back of your body** leading you through the space, even as you walk forward. You might also try walking backward. Notice the speed at which your body wishes to walk when led by this part of you, and let that happen; do not regulate yourself.

Let common phrases, metaphors, and experiences become visceral in your body. (*Start with one and then add others when it feels like enough time has passed for the metaphor to be explored. You might not get to all of them.*) Look back on it. Take a step back. Feel something at the back

of your mind. Who has your back? Imagine that you have just turned away after saying goodbye to someone but can still feel their presence behind you. Feel someone coming up behind you. Feel the hairs on your neck stand up. Feel your past, your ancestors, everyone who came before you, all the yous you have been. Notice whether you, or perhaps parts of you, resist moving all the way back. Notice any memories, texts, or feelings that come to mind.

Now, bring your awareness to **the front of your body** without losing **the back of your body** so that you feel both equally. You feel **the front of your body** without being pulled forward. You feel **the back of your body** without being drawn back. You are totally in the present. Notice whether your body is more receptive to one than the other, whether one is more familiar. Notice any memories, texts, or feelings that come to mind. Consider where in the dimension of **the front of your body** and **the back of your body** you feel the most personally present. Consider what difference it might make if you preached from that place.

4.  Text

Slowly read a text aloud, lingering in places that refer to specific parts of the body or seem to suggest these different dimensions. Let your whole body experience the text, moving as it pleases. Recognize words that could not be understood without a body. Here is Psalm 16:7–11, with body parts, senses, and possible dimensions italicized:

I bless the LORD who gives me counsel;

in the night also my *heart* instructs me.

I keep the LORD always *before* me;

because he is at my *right hand*, I shall *not be moved*.

Therefore my *heart* is glad, and my *soul* rejoices;

my *body* also rests secure.

For you do not *give me up* to Sheol,

or let your faithful one *see* the Pit.

You *show* me the path of life.

In your *presence* there is fullness of joy;

in your *right hand* are pleasures forevermore.

This is a sampling of exercises that I have found useful and that I hope can help preachers strengthen their ability to be personally present to their words and their people as they give them the gift of the gospel. I invite homileticians to adapt these exercises as needed and to develop their own. I recommend students be given a way to process their experience, whether through a class discussion immediately following the exercise or through a reflective journaling assignment.[34] It is not a small thing to be personally present to the words God uses to bring your people from death to life, and it is not a small thing for you to be "in this room" with your people as God does.

## 5. Christ for You and Me

My homiletical research and praxis are designed to help preachers proclaim the Gospel in such a way that Christ "is not simply 'Christ' but 'Christ for you and me,' and what we say about him and call him affect us" (Luther and Wengert 2016, p. 508). I am delighted to have dived into this treasure trove of Lutheran proclamation that resonates so richly and is adding new harmonies to my own sense of the importance of fostering the preacher's personal presence as they declare Christ's trustworthiness, that the promises of the gospel are promises "for you." I have only scratched the surface of what this tradition has to offer

homiletical conversations around the presence of the preacher and around performance studies more broadly.

The place of performance studies in homiletics is shifting. Some of the pillars of this subdiscipline have retired or are approaching retirement. Prime examples are Charles Bartow, who taught preachers to see the ink on the page of Scripture as a mere "arrested performance" that must be turned back into blood and live again through our embodied performance (Bartow 1997); Jana Childers, who recognized the psychophysical nature of preaching and taught preachers what actors know about the wisdom of the body in creativity and performance (Childers 2005, 2006); and Richard Ward, who urged homileticians to strengthen our theologies of communication, that we might bring voice training in from the margins and help our students "find their voices" with integrity (Ward 2008). Each has forged pathways of embodied preaching that gratefully embrace the learnings of other performative disciplines while thoughtfully developing performance studies in homiletics as a rich theological discipline in its own right. These pathways will be up to the next generations to prune and grow. Meanwhile, the Academy of Homiletics has been taking tangible steps to unmask its embedded racism, sexism, homophobia, etc., in ways that have revealed opportunities for its members to reevaluate some of our norms (cf. Winderweedle 2023). Lutheranism's particular for you-ness might be a gift on the journey to finding inclusive and expansive areas to grow roots that are nourished by a greater diversity of nutrients and bear fruit that tastes good and is accessible to more people in this next season.[35] Conversations around such "-isms" can quickly feel like law that puts to death that which we have devoted years of our lives learning. But as a tradition that prioritizes the explicit proclamation of the gospel, Lutheranism will never let us lose sight of the fact that law is never an end in itself and is never the end of the story. God is always working gospel, forgiveness, and new life, "for you." The crucial conversations unmasking bias and prejudice within homiletics are increasing the diversity of bodies and voices through which we experience the proclamation of the gospel and, thereby, are enriching our understanding of it. The new life that is rising up through this kind of preaching praxis can help the personal presence we foster in the pulpit be a presence that communicates Christ "for you" in such a way that, by the grace of God, no one feels excluded, but each individual person cannot help but respond "Amen."

**Funding:** This research received no external funding.

**Data Availability Statement:** No new data were created or analyzed in this study. Data sharing is not applicable to this article.

**Conflicts of Interest:** The author declares no conflict of interest.

## Notes

1.   I spent ten years (2012–2022) as a student, teaching assistant, and graduate instructor in Princeton Theological Seminary's speech department. For a historical overview of the Princeton school of speech through 1992, see (Bartow 1992). See also these more recent publications from Princeton Seminary speech faculty and a festschrift honoring the work of Charles Bartow: (Bartow 1997; Brothers 2014; Gross 2017; Childers and Schmit 2008).

2.   I was certified as a Designated Linklater Teacher in 2019. The methodology and basic outline of the exercises are found here: (Linklater 2006). Further resources can be found here: (Articles & Essays n.d.). Kristin Linklater developed this work with actors in mind; however, it has borne fruit in homiletics. See (Childers 1998; Hooke 2002, 2007, 2010, 2023; Gilmore 2022).

3.   For the purposes of this paper, I am assuming a neurotypical, able-bodied preacher and the mainstream United States culture in which I have always lived and worked. I recognize that context matters and that my description of presence does not apply to all people and communities.

4.   Charles Bartow, borrowing from performance studies, refers to the written text as an "arrested performance" that cannot be understood as an end in itself, but must be performed, played by the human body and voice. (Bartow 1997, p. 64)

5.   The body's vital contribution to communication can be witnessed in the everyday use of emojis and memes. These options allow for more holistic and specific expression than words alone.

6.   By filling in gaps, I mean that the preacher who is not personally present to the words does not communicate all of the information normally manifested by the body and voice. The listener is left to imagine the meaning that would be communicated through

these means. By deciphering contradictions, I mean that the body and voice may actually contradict the words. The words suggest freedom and abundance, but the body and voice suggest constraint and constriction. The words say "grace" and "mercy," but the body and voice communicate as one who is not experiencing them—so, are they real?

7   I acknowledge that not all of these things are possible in every congregation, especially larger congregations that place the pulpit far away from the congregation or use stage lights that inhibit seeing the congregation. This use of space may suggest a theology of preaching that does not prioritize personal presence with the congregation and the participation of the congregation in preaching. Large congregations, however, are a minority in the United States. Most congregations are small, so that congregation size is not a barrier to these things becoming realities. Small congregations worshipping in large buildings might be encouraged to sit further forward and closer together in the sanctuary or to worship in an alternative room in the building better suited to personal presence and congregational participation (Wingfield 2022; Earls 2021; Potter 2017).

8   "Indicate" is a common term in the theater used to refer to an actor who is attempting to *show* what their character is feeling or doing without actually feeling or doing it. An audience generally struggles to suspend disbelief with an actor who is indicating because they are distracted by the fake, inauthentic, or overacted performance. A preacher who indicated a relationship with a congregation that did not actually exist would probably also be experienced as fake, inauthentic, and "acting," in the worst sense of the word (e.g., pretending, lying).

9   Lutheran theologian Marit Trelstad notes that within the theological use of the law, one can discern within Luther's writings both "descriptive" and "performative" functions. She acknowledges the usefulness of the descriptive function that reveals sin by telling the truth "that is necessary for accountability, ethics, and liberation." She asks that Lutherans not accept the "performative" function, however, due to the way it displays abusive images of God as one who terrorizes, humiliates, and even "beats us up." Trelstad suggests that this language may be a result of Luther's family background and a "reentrenchment into the very grounds he struggled to escape." These images do not correspond to the God who revealed Godself in Jesus Christ, instead promoting a highly problematic relationship between God and humanity that violently impacts relationships between human beings (Trelstad 2015, pp. 216–21).Lutheran theologian Mary Streufert points to traditional interpretations of the cross with a similar concern about the implications of violence: "If our prime hermeneutic to understand the cross of Christ is through violent atonement, then violence can become our image and what we imitate." Streufert offers a hermeneutic of maternal sacrifice, a "life-for-life model" that "turns Christians more strongly toward Jesus' life as a potential locus of redemption." While this hermeneutic is beyond the scope of this paper, I recognize the violent abuse that results from misapplications of atonement theories and non-contextually sensitive applications of law and gospel (Streufert 2006, pp. 65, 74).

10   There is much discussion surrounding the third use of the law. Wengert explains that it was added later to the first two uses by Melanchthon as a way to clearly separate Lutherans from antinomians and to help distinguish law from gospel. All commands, even those given particularly to Christians (e.g., Jn 13:34), are law and should not be confused with gospel, which is pure and unconditional promise. When this third use is preached well, Wengert proposes that it is heard not as a "condemning command but rather enticing invitation" from a good, loving, and gracious father to his beloved children. Using a memorable image, Wengert proposes that "unless people come out of church whistling," the preacher is not preaching this use of the law and is instead burdening people with "one more thing to do to get right with God" (Wengert 2013, pp. 31, 37–40).

11   Or, in alignment with Trelstad, and with the understanding that the context is a worshipping congregation rooted in the gospel, Lutheran preachers might ask, "How God is working gospel-law-gospel on them and their people?" (Trelstad 2015, p. 209).

12   Because of this active way in which God's Word works law and gospel on the hearers, theologian Gerhard Forde advises preachers to preach in a correspondingly active manner, to do what they hear happening in the text as they listen with this law and gospel hermeneutic: "Preaching is doing the text to the hearers. *Doing* the text, not merely explaining it (though that will be involved), not merely exegeting the text (though that is presupposed and indispensable), not merely describing or prescribing what Christians are supposed to do (though that will no doubt result). Preaching in a sacramental fashion is *doing* to the hearers what the text authorizes you to do to them." What the text authorizes, he proposes, is that the preacher aims to do the "same thing in the present to the assembled hearers" as the text "actually *did*" (Forde 2017, pp. 91, 94).

13   I recognize that some readers might have concerns here and throughout the paper about the implications of this personal focus and its potential for an inwardness that forgets or ignores the neighbor. The personal and the social, however, are not mutually exclusive. The gospel is personally experienced in relationship in preaching (among and between preacher, congregation, and God in Christ by the power of the Holy Spirit). Moreover, the transforming power of the gospel is ideally what inspires, undergirds, nourishes, and guides Christians' good work for justice in the world. Without attention given to the personal, good work may become disconnected from that which inspired it in the first place and be more difficult to sustain. Preaching law and gospel holds the potential both to honestly reveal that which has been done and left undone in this world and to offer forgiveness and a new identity oriented toward trust in Christ that frees Christians to turn toward their neighbor for whom Christ died.

14   Lest one be concerned about the power this gives the preacher, no such claim is made about anything else that a preacher might choose to proclaim.

15   This is not to negate the importance of acting on the implications of our faith but to properly label that action "law" and not "gospel," not salvific, not having anything to do with our standing before God. The gospel is solely God's action and not our action. If that distinction is not clear, the message of the gospel may be perplexing. Comparing preaching to the sacraments,

Forde expresses concern that "what is said in the sermon is all too often quite at odds with what we do in the sacraments. If we give unconditionally in the sacraments we are likely to take it back or put conditions on it in the sermon and leave our people completely confused. We are likely to imply in our preaching that the gift is not really what it is cracked up to be so now they better get *really* serious" (Forde 2017, p. 89). This concern echoes throughout this tradition that holds fast to justification. As homiletician Shauna Hannan notes, Article Four of the Augsburg Confession, "Concerning Justification," is the article by which the Lutheran church stands and falls (Hannan 2022, p. 45).

16    Meuser goes so far as to say that, for Luther, "nothing except Christ is to be preached—Christ as Savior, the one in whom God shows [God's] own face, in whom God has done a once-for-all deed and spoken a one-for-all definitive word to the world" (Meuser 2006, p. 138).

17    Meuser expands upon Luther's prioritization of the spoken word, quoting him saying, "The devil cares nothing about the written word, but where one speaks and preaches it, there he takes to flight" (Meuser 2006, p. 141). A particular person standing before others and proclaiming Christ to particular listeners holds a power that the written word alone does not have.

18    As Wengert puts it: "Yet apostles—and pastors and preachers are all apostles—have only one office, to bear witness to Christ crucified and risen again for the life of the world. That is all anyone has: God's weakness in Christ *is* the believer's strength" (Wengert 2013, p. 53; emphasis original).

19    Forde exhorts preachers dauntlessly to declare the sacramental nature of preaching: "The preacher has to have the audacity to believe that at the very moment of preaching is itself the sacrament, the audacity to claim that from all eternity God has been preparing for just this very moment and thus to say, 'Here it is, it is for you!' The preaching itself is the treasure, the sacramental moment" (Forde 2017, p. 97).

20    One of the major differences in the experience of reading versus being spoken to is that the latter engages more senses, which can create a richer and more holistic experience. Kolb notes Luther's appreciation of the ways in which God graciously communicates God's Word by addressing all human senses: "In 1538 [Luther] exulted that God graciously addresses the five human senses. Through the hand and tongue of the minister of the gospel God is at work. In baptism there is an oral word and a pourer, in the sacrament an oral word and a feeder, in preaching an oral word and a speaker, as in absolution" (Kolb 2009, p. 136).

21    Some of my teachers have said things in jest like, "What a ridiculous thing for you to be doing with your serious adult time…" or, "Are you breathing? Sometimes we stop breathing when someone asks us to do something strange like this." These kinds of comments acknowledge discomfort in a way that gives people permission to laugh about it with the teacher. The laughter helps to break through the tension of the discomfort to a playful and fruitful freedom on the other side.

22    Students who want more intellectual verification that they are not wasting their time with these kinds of exercises can be pointed to embodied cognition science. For instance, see (Hrach 2021).

23    The professor who does the exercise with the students sees less of what the students are experiencing, and thus has less awareness of how an exercise is landing on a particular group of students. Leading while doing can also give a sometimes unhelpful impression of how the students "should" be experiencing a particular exercise. In more self-conscious groups, however, the professor's participation can create room for students to engage with less self-consciousness because there is less attention on them from the professor.

24    Consider how the mere mention of delicious food can make one's mouth water, and how thinking about a past incident, particularly an intense one, can bring all those heart-racing feelings physically rushing back to the body. The mind–body connection is so strong that the mind can move the body internally without external movement (and external movement can change the mind).

25    Due to the body's connection with emotion, students sometimes experience strong feelings while practicing the exercises. This is generally not an issue and even a positive thing; feeling feelings is part of being human and a vital part of preaching (Childers 2006, pp. 229–30). If the experience is overwhelming to the student, however, the professor may provide contextually appropriate options depending on the nature of the group and the student. In a small group that has developed trust, the student may wish to share what is happening and then dive back in. In other instances, the student may wish to step to the side and journal while the rest of the class keeps going.

26    If students are synchronously online, the professor may invite them to have their cameras on and to be visible so that they can see how the exercise is landing and whether more or less time is needed for the exploration of movement. If the professor is recording for asynchronous students, they will want to leave space in the recording for that exploration.

27    This step is a common way to begin Linklater warmups.

28    This exercise was developed by internationally recognized voice teacher Kristin Linklater. I have come to know it as I do through repetition. I am writing from my own memory and with no authoritative written source. Undoubtedly, were Linklater to write this down, she would do so differently. The exercise is briefly described by homiletician Ruthanna Hooke here: (Hooke 2002, p. 13).

29    If students are synchronously online, students should instead be invited to have their cameras on and to be visible to their peers. Professors may wish to change the language "in this room" to "in this space" or another word that gives a sense that all are together even in their different rooms. I have recorded this exercise for asynchronous students and invited them to close their eyes and imagine the people in their preaching context during the "with all of you" section.

[30]    It is common for people to stop breathing or to breathe shallowly when they feel vulnerable, which is often the case when being asked to simply "be" while being seen by other people (rather than doing something that can draw their attention away from your being to the activity) in the way this exercise invites. Holding one's breath is a way of disappearing from the room, of removing one's presence.

[31]    Numbers 1-6 are a common sequence of stretches used by Designated Linklater Teacher Andrea Haring at the beginning of warmups. I have come to know it as I do through repetition. I am writing from my own memory and with no authoritative written source. Undoubtedly, were Haring to write this down, she would do so differently. Haring is the Executive Director of the Linklater Center for Voice and Language (see Haring 2024) and was my primary teacher on the path to becoming a Designated Linklater Teacher.If students are synchronously online, students may be invited to have their cameras on and to be visible, in addition to unmuting themselves, so that when the praying begins, there is a greater sense of praying together in unison rather than by themselves in their rooms. If recording for asynchronous students, the professor will want to leave space in the recording for the students to repeat each phrase.

[32]    This exercise was developed by movement teacher Merry Conway (see Conway 2011). I have come to know it as I do through repetition in her workshops. I am writing from my own memory and with no authoritative written source. Undoubtedly, were Conway to write this down, she would do so differently. I have added a few preaching reflections that she would not include, as her primary audience is actors.I let students know beforehand that they will want to dress in comfortable clothing that is easy to move around in for this exercise. If they are synchronously online, I invite them to create as much physical space for themselves in their room as possible for exploration. While I ask them to have their cameras on so that I can see how the exercise is landing and whether more or less time is needed for exploration, I acknowledge that they will likely be moving on and off screen throughout the exercise and that this is fine. I have not led this exercise asynchronously and question whether it would work well in that modality. The time spent in each section varies significantly with each group, making prerecording difficult. Additionally, the longer length of this exercise might be challenging for a student to sustain without the structure of a synchronous class.

[33]    It is common to spend a lot of time here in the United States, where value is assigned according to how busy and productive we are. We tend to notice our physical sensations only when something is "wrong," such as when we are hungry, hot, cold, in pain, or nervous (manifested by held or shallow breath, an upset stomach, wobbly knees, sweaty palms, a racing heart, a dry mouth, or the like).

[34]    Class discussions can provide space for students to recognize that they are not the only person having a particular experience or the only person with a particular question. It can also reveal the very different ways that exercises can be experienced (e.g., one feels crushed by gravity's weight, while another feels empowered by the way it grounds them). The vulnerability this kind of sharing requires often deepens relationships within the group, which is beneficial in a preaching classroom where students provide feedback on one another's sermons.

[35]    I recognize that this sentence might sound odd. While Lutheranism is globally diverse, the Evangelical Lutheran Church in America is the whitest denomination in the United States (Allende 2021). The student body of my institution is representative of that reality (though, crucially, it is not without diverse representation in other ways). Still, the Lutheran recognition of the law's value and the need for it to be proclaimed, while refusing to let it overwhelm or deny the forgiveness and new life God is working through the proclamation of the gospel in the present tense, can encourage us to openly acknowledge the truth about the state of injustice in our guild and participate with hope in the redemption and new life God is bringing. Lutheranism's for you-ness compels us to ensure as best we can that each particular "you" at our annual meetings, in our classrooms, and in our students' ministries experiences the gospel "for you" and not "for someone else."

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
