# Peer review of "“Christ for You and Me”: A Lutheran Theology of Proclamation and the Presence of the Preacher"

_religions, doi:10.3390/rel15030272_

Round 1

Reviewer 1 Report

Comments and Suggestions for Authors

Excellent paper outlining a theology of homiletical presence and a practical guide to cultivating an embodied presence. The only difficulty in this paper is that there is a generalization of "the Lutheran tradition" that is justified by an appeal to a very narrow slice of white male conservative law/gospel oriented Lutheran theologians--Forde, Wengert, Bayer. I request that the author also address the ground breaking work done by women scholars of Luther, as they have advanced new ideas that are not dominated by the law/gospel model as well as being at the forefront of theologies of embodiment: Marit Trelstad (who challenges the law/gospel model for its abusive implications); Mary Streufert; Kirsi Stjerna (recent book on Lutheran Theology); the Danes--Anna Vind and Else Marie Wiberg Pedersen who also challenge the law/gospel dialectic on the basis of a robust doctrine of creation; Candace Kohli. 

Author Response

Thank you very much for taking the time to review this paper. I am especially grateful for the specific names and resources you offer to address women scholars to provide a more expansive view of the Lutheran tradition. I agree with your assessment. In response, I have engaged a chapter by Marit Trelstad in which she speaks directly to law and gospel (lines 156-158; 160-163; footnotes 9 and 11), Kirsi Stjerna’s recent book on Lutheran theology (lines 143-144; 153-155; 163-166), and a chapter by Mary Streufert (footnote 9) in which she speaks with concern about the violence of traditional interpretations of the cross.

Thank you again. I look forward to reading all of the voices you have named.

Reviewer 2 Report

Comments and Suggestions for Authors

The author fulfills their stated purpose and method. This article should be published.

Here are some curiosities that I suspect will parallel how this essay might be received in broader discourse within the realms in which I work. 

I am curious why the particularities of bodies and social systems that shape so much of our bodily presence in the world and church are left to the conclusion. 

I am curious about how to utilize some of these exercises in an online or multimodal classroom or space. 

As someone who comes from a tradition (non-Lutheran) that a) has emphasized personal presence albeit with different theological rationale and b) not unrelatedly has also emphasized personal piety above social dimensions of encountering the Gospel, I am curious about how this personal presence functions in broader social bodies to situate the preacher in relationship to the struggle for justice in the world.  

I was initially curious as to why the author turned to Euro-American discourse as the center of their constructive work. There was a helpful piece in 689-691 that possibly hedges against this critique and provides a sensible rationale for starting with the familiar because it might lead to broader connections. I am curious what it might look like for the author to briefly and specifically envision how attending to predominantly Euro-American Lutheran voices can invite more diverse connections.

Thanks for an intriguing essay!

Author Response

Thank you very much for taking the time to review this paper and for the many curiosities you offer that will keep me pondering for a long time to come. In footnotes 26, 29, 31, and 32, I address your curiosity about using the exercises in different modalities. In footnote 13, I begin to address your curiosity about personal presence leading to an emphasis on piety above the social dimensions of the gospel. In footnote 36, I begin to address your curiosity about Euro-American Lutheran voices inviting more diverse connections. While I do not address your curiosity about leaving any discussion about our bodily presence in the world and the church to the end, I am grateful for this note and will keep it with me as I move forward. Thank you again for your time and expertise.
